# An Optimization Perspective on Realizing Backdoor Injection Attacks on Deep Neural Networks in Hardware

## Abstract

State-of-the-art deep neural networks (DNNs) have been proven to be vulnerable to adversarial manipulation and backdoor attacks. Backdoored models deviate from expected behavior on inputs with predefined triggers while retaining performance on clean data. Recent works focus on software simulation of backdoor injection during the inference phase by modifying network weights, which we find often unrealistic in practice due to the hardware restriction such as bit allocation in memory. In contrast, in this work, we investigate the viability of backdoor injection attacks in real-life deployments of DNNs on hardware and address such practical issues in hardware implementation from a novel optimization perspective. We are motivated by the fact that the vulnerable memory locations are very rare, device-specific, and sparsely distributed. Consequently, we propose a novel network training algorithm based on constrained optimization for realistic backdoor injection attack in hardware. By modifying parameters uniformly across the convolutional and fully-connected layers as well as optimizing the trigger pattern together, we achieve the state-of-the-art attack performance with fewer bit flips. For instance, our method on a hardware-deployed ResNet-20 model trained on CIFAR-10 can achieve over 91% test accuracy and 94% attack success rate by flipping only 10 bits out of 2.2 million bits.

## 1 Introduction

DNN models are known for their powerful feature extraction, representation and classification capabilities. However, the large number of parameters and need for a large training dataset makes it hard to interpret the behavior of these models. The fact that increasing number of security critical systems rely on DNN models in real-world deployments raises numerous robustness and security questions. Indeed, DNN models have been shown to be vulnerable against imperceivable perturbations to input samples which can be misclassified by manipulating the network weights (Szegedy et al., 2013; Goodfellow et al., 2014; Nguyen et al., 2015).

Emboldened by recent physical fault injection attacks, *e.g.,* Rowhammer, an alternative approach was proposed that directly targets the model when it is loaded into memory. The advantage of this attack approach is that it can remain completely stealthy since the model is only modified in real-time while running in memory and no modification to inputs is required. Recently, Hong et al. (2019); Yao et al. (2020) showed that flipping a few bits in DNN model weights in memory while succeeding in achieving misclassification, has the side-effect of significant accuracy drops which renders the model useless. Liu et al. (2017a) and Bai et al. (2021) addressed this problem by tweaking only a minimum number of model weights that makes a DNN model misclassify a chosen input to a target label. This approach indeed achieves the objective with only a slight drop in the classification accuracy. Nevertheless, whether a practical attack such as injecting a backdoor to DNNs can indeed be realized in a scalable and stealthy manner using Rowhammer in hardware is still an open question. Earlier approaches assume that Rowhammer can flip any bits in the memory. This is far from what

we observe in reality: only a small fraction of the memory cells are vulnerable. This motivates us to reconsider the backdoor injection process under new constraints, including the training algorithms.

**Contributions.** In this paper we propose a novel algorithm based on constrained optimization that can identify vulnerable bits in the memory for deep learning model to create a backdoor. To this end, we introduce a new fine-tuning process that takes into account the stringent constraints on faults obtained using Rowhammer characterization experiments. To show the practicality of our approach, by fine-tuning the model we first identify the necessary bit flips to implement the backdoor. Using the Rowhammer attack we inject the backdoor in a DRAM setup to the deployed model. Once the backdoor is injected, a DNN model will misclassify any sample with the trigger pattern to the target class without degrading the overall accuracy on clean data. This result shows that indeed real-life deployments are under threat from backdoor injection attacks with potentially adverse affects to the well being of the society. More work needs to be done to secure deployed models from fault injection attacks used for every day tasks by end users.

## 2 BACKGROUND AND RELATED WORKS

### 2.1 BACKDOOR ATTACKS

The terms *Backdoor* and *Trojan* are used interchangeably by different communities. Here we use Backdoor for consistency. In DNN models, we define a *Backdoor* as a hidden feature that causes a change in the behavior that is triggered only by the existence of special type of inputs. In the literature, backdooring is applied with either benevolent intents, such as watermarking the DNN models, or with malicious intents, as a *Trojan* to attack the models. In this work, we focus on *Backdoor* as a type of *Trojan* that is exploited by an attacker to cause targeted misclassification.

### 2.2 ATTACKS ON DEEP LEARNING MODELS

DNN model weights have been shown to be vulnerable to Rowhammer attacks (Hong et al., 2019) causing accuracy degradation even after the use of quantization (Yao et al., 2020) as a defense. A binary integer programming approach was proposed (Bai et al., 2021) to find the minimum number of bit changes required to manipulate the model for one specific example. Further Rakin et al. (2020b) showed that samples from single or multiple classes can be misclassified to a target class using Rowhammer. However, this attack is not in the scope of backdoor attacks since it does not use a trigger. Earlier works by Liu et al. (2017b) and Gu et al. (2017) demonstrated that backdoor attacks pose a threat to the DNN model supply chain which can be exploited by poisoning data sets (Gu et al., 2017), poisoning the training code (Bagdasaryan & Shmatikov, 2020), modifying network connections or modifying hardware (Clements & Lao, 2018; Venceslai et al., 2020). Pang et al. (2020) proposed mutual optimization of adversarial input and the poisoned models. Recently, Rakin et al. (2020a) showed that backdoor attacks can be implemented by changing only a small number of weight parameters assuming any bit in the memory can be flipped. However, the practicality of software-based backdoor injection attacks during the inference phase is still an open question due to the practical constraints that are overlooked in previous works.

### 2.3 ROWHAMMER ATTACK

As memories are becoming more compact and memory cells are getting closer and closer, the boundaries between the DRAM rows do not provide sufficient isolation from electrical interference. The data is encoded in the form of voltage levels inside a capacitor which leaks charge over time. Thus they need to be refreshed after every 64ms to keep the data intact. However, if there is an attacker residing in a nearby DRAM row, although in a completely isolated process, the attacker can cause a faster leakage in the victim row by just accessing his own memory space repeatedly (hammering). Rowhammer is a software induced fault attack first introduced by Kim et al. (2014). Recently, Frigo et al. (2020) and de Ridder et al. (2021) have shown that more than 80% of the DRAM chips in the market are vulnerable to the Rowhammer attack including DDR4 chips having Target Row Refresh (TRR) mitigation. The Error Correcting Codes (ECC) mitigation has also been bypassed in Cojocar et al. (2019). Rowhammer is a great threat to shared cloud environments (Cojocar et al., 2020; Xiao et al., 2016) as it can be launched across virtual machine (VM) boundaries and even

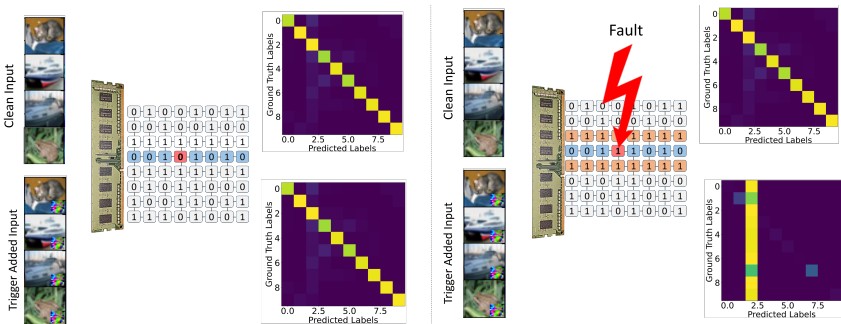

Figure 1: Clean Model (left) vs. Backdoored Model (right) behavior with clean inputs (top) and trigger added inputs (bottom). Note the fault injection (right) to model in memory device changes behavior of classifier as shown by the confusion matrix (bottom, right).

remotely through JavaScript. More recently, Qazi et al. (2021) have shown a combined effect of more than two aggressor rows to induce bit flips in recent generations of DRAM chips. All existing Rowhammer defenses including TRR, ECC, detection using Hardware Performance Counters and changing the refresh rate can not fully prevent Rowhammer attack (Gruss et al., 2018; Frigo et al., 2020). The only requirement of the Rowhammer attack is that the attacker and the victim share the same DRAM chip, vulnerable to the Rowhammer attack.

## 3 METHODOLOGY

### 3.1 THREAT MODELS

Same as in earlier works (Gu et al., 2017; Rakin et al., 2020a; Liu et al., 2017b; Hong et al., 2019; Yao et al., 2020), we assume that the attacker

- only knows the model architecture and model parameters;
- has a small percentage of the unseen test data set;
- does not have access to the training hyperparameters, or the training dataset;
- is involved only after the model deployment in a cloud server, and therefore does not need to modify the software supply chain;
- resides in the same physical memory as the target model;
- has no more than regular user privileges (no root access).

Such threat models are well motivated in shared cloud instances targeting a co-located host running the model, and in sandboxed browsers targeting a model residing in the memory of the host machine (Cojocar et al., 2020; Xiao et al., 2016; de Ridder et al., 2021).

To better understand our attacker, we illustrate an example in Fig. 1. The attacker works as follows:

1. *Offline Attack Phase:* By studying the model parameters and the memory, the attacker generates a trigger pattern and determines the vulnerable bits in the target model.
2. *Dynamic Trigger Injection:* After the target model is loaded into the memory, using Rowhammer the attacker flips the target bits by only accessing its own data that resides in the neighboring rows of the weight matrices in the DRAM.
3. *Misclassification:* After the backdoor is inserted, the model will misclassify the trigger-added input to the target class. The misclassification will persist until the backdoored model is unloaded from the memory. Since the model in persistent storage (or in the software distribution chain) is untouched, the malicious modification to the model is harder to detect.

### 3.2 OFFLINE ATTACK PHASE

In the offline phase of the attack, we optimize the trigger pattern and the bit-flip locations in the weight matrices. To do so, we first extract the profile of vulnerable bits in the DRAM and then

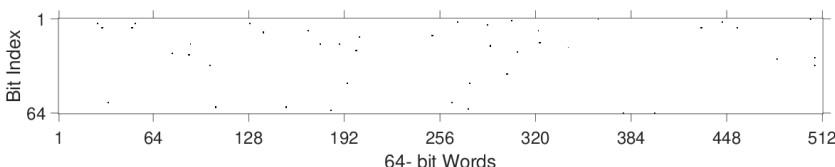

Figure 2: Bit flips in one of the 4KB pages showing the sparsity of the actual bit flips.

train the backdoor model with new constraints. Note that this phase is independent on the hardware specification, and the learned model can be used freely in any DRAM.

### 3.2.1 ROWHAMMER PROFILING

Rowhammer profiling is a process of finding vulnerable memory addresses in the DRAM. This process can be performed offline before the victim starts running. We have used the Hammertime tool [1] by Tatar et al. (2018) to profile our DRAM with double-sided Rowhammer. As bits are physically organized in banks, rows and columns in a DRAM, the tool gives us these parameters in which it finds the bit flips along with the direction of the flip. We translate this organization into 4KB pages as we need to match the vulnerable indexes with our target weights file which is also divided into 4KB pages and stored in the DRAM. Among all the available Rowhammer tools, Hammertime gives the most number of bit flips as shown in (Tatar et al., 2018). In practice, we observe that only **0.036%** of the total cells in the 128MB buffer have bit flips, as illustrated in Fig. 2. Hence, assuming a specific sequence of bit flips within a page like in previous research is highly unrealistic.

### 3.2.2 CONSTRAINED FINE-TUNING WITH BIT REDUCTION

In contrast to previous works such as (Liu et al., 2017b; Rakin et al., 2020a), in this paper we propose a novel *joint* learning framework based on constrained optimization to learn the network weight modification pattern as well as the data trigger pattern simultaneously. Also different from the literature, we do not rely on the last layer only to find vulnerable weights; instead we achieve a wider attack surface on the model with constraints placed on the number and location of faults.

To preserve the performance of the networks on clean data, given a collection of test samples $\{x_i\}$ and their corresponding class labels $\{y_i\}$, we propose optimizing the following objective:

$$\min_{\Delta\theta\in\Delta\Theta} \max_{\|\Delta x\|_\infty\leq\epsilon} F(\Delta\theta, \Delta x)$$
$$= \sum_i \left[ (1-\alpha)\cdot\ell\Big(f(x_i, \theta+\Delta\theta), y_i\Big) + \alpha\cdot\ell\Big(f(x_i+\Delta x, \theta+\Delta\theta), \tilde{y}\Big) \right], \quad (1)$$

where $\Delta\theta, \Delta x$ denote the weight modification pattern and the data trigger pattern, $\tilde{y}$ denotes the target label, $\ell$ denotes a loss function, $f$ denotes the network parameterized by $\theta$ originally, $\alpha \in [0,1]$ denotes a predefined trade-off parameter to balance the losses on clean data and triggered data, and $\epsilon \geq 0$ denotes another predefined parameter to control the trigger pattern. Note that $\Delta\Theta$ denotes a feasible solution space that is restricted by the hardware implementation requirements. Specifically,

> **Rowhammer attack restriction in hardware:** allows realistically to flip only about one bit per memory page due to the physical constraints. Since the potentially vulnerable memory cells in the DRAM are sparse (See Fig. 2.), we cannot flip multiple bits at the targeted page. Such a restriction forms the feasible solution space $\Delta\Theta$ in learning.

To solve the constrained optimization problem defined in Eq. 1, we also propose a novel learning algorithm as listed in Alg. 1 that consists of the following 4 steps:

**Step 1. Learning data trigger pattern** $\Delta x$**.** The goal of this step is to learn a trigger that can activate the neurons related to the target label $\tilde{y}$ to fool the network. Trigger pattern generation starts with

---

[1]For the implementation details of Rowhammer profiling, we refer readers to Hammertime repository. https://github.com/vusec/hammertime

---

**Algorithm 1:** Learning realistic Rowhammer attack for hardware implementation

---

**Input:** A DNN model with weights $\theta$, number of bits $N_{flip}$ that are allowed to be flipped in the memory, objective $F$, parameter $\epsilon$, learning rate $\eta$, and maximum number of iterations $T$
**Output:** Backdoored model $\theta^*$ and data trigger pattern $\Delta x^*$

---

$\Delta \theta^* \leftarrow \emptyset, \Delta x^* \leftarrow \emptyset$;
**for** $t \in [T]$ **do**
    **if** *update the trigger == true* **then**
        $\Delta x^* \leftarrow \Delta x^* + \epsilon \cdot \mathrm{sgn}(\nabla_{\Delta x} F(\Delta \theta^*, \Delta x^*))$;
    **end**
    $\mathcal{M} \leftarrow Group\_Sort\_Select(|\nabla_{\Delta \theta} F(\Delta \theta^*, \Delta x^*)|, N_{flip}, 'descending')$;
    $\Delta \theta^* \leftarrow \Delta \theta^* - \eta \cdot [\nabla_{\Delta \theta} F(\Delta \theta^*, \Delta x^*)]_{\mathcal{M}}$;
    **if** *bit reduction == true* **then**
        $\theta^* \leftarrow \mathrm{Floor}((\theta + \Delta \theta^*) \oplus \theta) \oplus \theta$;
    **end**
**end**
**return** $\theta^*, \Delta x^*$

---

an initial trigger mask. Then we simply use the Fast Gradient Sign Method (FGSM) (Goodfellow et al., 2014) to learn the trigger pattern. The update rule is defined as

$$\Delta x = \Delta x^* + \epsilon \cdot \mathrm{sgn}(\nabla_{\Delta x} F(\Delta \theta^*, \Delta x^*)), \tag{2}$$

where $\Delta \theta^*, \Delta x^*$ denote the current solutions for the two variables, $\nabla$ denotes the gradient operator, and $\mathrm{sgn}$ denotes the signum function.

**Step 2. Locating vulnerable weights.** Now given a number of bits that need to be flipped, $N_{flip}$, which should be no bigger than the page number in the memory, in this step our goals are:

• C1. Locating one weight per bit flip towards minimizing our objective in Eq. 1 significantly;
• C2. No co-occurrence in the same memory page among the flipped bits.

In fact, these two constraints define a combinatorial optimization problem (that is NP-hard, in general (Papadimitriou & Steiglitz, 1998)) over a huge space of network parameters. Therefore, solving the problem in Eq. 1 with such constraints exactly will be extremely challenging.

Instead, we relax our learning problem by ignoring Constraint (C2) and locate vulnerable weights heuristically. Recall that when a DNN model is fed into the memory, the network weights are loaded sequentially page-by-page where each page is fixed-length and stored contiguously. Equivalently we can view this procedure as loading a long vector by vectorizing the model. Therefore, to locate vulnerable weights, we simply divide the network weight vector into $N_{flip}$ groups as equally as possible, and rank the weights per group based on the absolute values in the gradient over $\Delta \theta$, *i.e.,* $|\nabla_{\Delta \theta} F|$ where $|\cdot|$ denotes the entry-wise absolute operator, in descending order. The top-1 weight per group will be identified as a vulnerable weight.

Because empirically the number of pages in the memory that is occupied by the network model is much larger than $N_{flip}$, based on the pigeonhole principle the co-occurrence probability among flipped bits is very low. Actually we have not observed such a case in our experiments.

**Step 3. Adversarial fine-tuning.** Now given a collection of located vulnerable weights, denoted by $\mathcal{M}$, we only need to update these weights in backpropagation as follows:

$$\Delta \theta = \Delta \theta^* - \eta \cdot [\nabla_{\Delta \theta} F(\Delta \theta^*, \Delta x^*)]_{\mathcal{M}}, \tag{3}$$

where $[\cdot]_{\mathcal{M}}$ denotes a masking function that returns the gradients for the weights in $\mathcal{M}$, otherwise 0's, and $\eta \geq 0$ denotes a learning rate.

**Step 4. Bit reduction.** To meet the physical constraints of the Rowhammer attack, the final part of our attack procedure requires bit reduction. Rowhammer can only flip a very low number of bits in a 4KB memory page and more than one faulty memory cell almost never coexists within a byte. Therefore, we define a bit reduction function as $\mathrm{Floor}(\theta \oplus \theta^*)$, where $\oplus$ denotes the bit-wise summation, and function $\mathrm{Floor}$ rounds down the number by keeping the most significant nonzero bit

only. For instance, letting $\theta = 1101_2$ and $\theta^* = 1010_2$, then $\text{Floor}(\theta \oplus \theta^*) = \text{Floor}(0111_2) = 100_2$. In this way, we assures that only one bit is modified in a selected weight while maintaining its change direction and amount as much as possible.

### 3.3 ONLINE ATTACK PHASE: FLIPPING BITS IN THE DEPLOYED MODEL IN MEMORY

When we access a file from the secondary storage, it is first loaded into the DRAM and when we close the file, the operating system (OS) does not delete the file from the DRAM to make the subsequent accesses faster. However, it shows that space as free to the user and utilizes it as page cache. If the file is modified, the OS sets the dirty bit of that modified page and it is written back according to the configured write back policy. Otherwise, the file remains cached unless evicted by some other process or file. As Rowhammer is capable of flipping bits in DRAM, we can use it in the online attack phase to flip the weights of the DNN file as it is loaded in the page cache. The weights file is divided into pages and stored in the page cache. We can flip our target bits as identified by the backdoored parameters $\theta^*$, in Section 3.2. The OS does not detect this change as it is directly made in the hardware by a completely isolated process and it keeps providing the page cached modified copy to the victim on subsequent accesses. Thus the attack remains stealthy.

### 3.4 WEIGHT QUANTIZATION

The weights are stored as $N_q$-bit quantized values in the memory as implemented in NVIDIA TensorRT (Migacz, 2017), a high performance DNN optimizer for deployment that utilizes quantized weights (NVIDIA). Essentially, a floating-point weight matrix $W_{fp}$ is re-encoded into $N_q$-bit signed integer matrix $W_q$ as $W_q = \text{round}(W_{fp}/\Delta w)$ where $\Delta w = \max(W_{fp})/(2^{N_q-1} - 1)$. In our experiments, we use 8-bit quantization and the weights are stored in two's complement form.

## 4 EXPERIMENTS

### 4.1 EXPERIMENTAL SETUP AND EVALUATION METRICS

To demonstrate the viability of our attack in the real-world, we implemented it on an 8-bit quantized ResNet-18 model trained on CIFAR-10 using PyTorch v1.8.1 library. The clean model weights that are trained on CIFAR-10 are taken from (Rakin et al., 2020a) for ResNet-18 and from (Idelbayev)(580 stars on GitHub) for other ResNet models. Moreover, we experimented on larger versions of ResNet models, such as ResNet50, trained on ImageNet data set. For the models that are trained on ImageNet, we use pretrained models of Torchvision library (9.1K stars on GitHub) which is downloaded 28 million times until now (Pepy.tech., Retrieved September 23, 2021). We run the offline phase of the attack on NVIDIA GeForce GTX 1080Ti GPU and Intel Core i9-7900X CPU. Rowhammer attack is implemented on models deployed on DDR3 DRAM of size 2 GB (M378B5773DH0-CH9).

We compare our approach with BadNet (Gu et al., 2017), and TBT (Rakin et al., 2020a) as well as fine tuning (FT) the last layer. We also include the output of our Constrained Fine Tuning (CFT) without bit reduction in Table 2 for comparison. For the results on software, we keep all the bit flips in the weight parameters assuming they are all viable. In the hardware results, we keep the bits that are possible to be flipped by Rowhammer and exclude the others. We use 128 images from the unseen test data set for all the experiments in CIFAR-10. Test Accuracy and Attack Success Rate metrics are calculated on unseen test data set of 10K images. In all experiments we used $\alpha = 0.5$ for Algorithm 1. The trigger masks are initialized as black square on the bottom right corner of the clean images with sizes 10x10 and 73x73 on CIFAR-10 and ImageNet respectively. $\epsilon$ in Eq. 2 is chosen as 0.001. For the ImageNet experiments, we use 1024 images from the unseen test data set to cover all 1000 classes. Test Accuracy and Attack Success Rate metrics are calculated on unseen test data set of 50K.

**Number of Bit Flips** ($N_{flip}$)  As in (Yao et al., 2020; Hong et al., 2019; Rakin et al., 2020a; Bai et al., 2021), the first metric we use to evaluate our method is $N_{flip}$ which indicates how many bits are different in the new version of the model in total. The $N_{flip}$ has to be as low as possible because only a limited number of bit locations are vulnerable to the Rowhammer attack in DRAM.

As the $N_{flip}$ increases, the probability of finding a right match of vulnerable bit offsets decreases. $N_{flip}$ is calculated as $N_{flip} = \sum_{l=1}^{L} D(\theta^{[l]}, \theta^{*[l]})$, where $D$ is the hamming distance between the parameters $\theta^{[l]}$ and $\theta^{*[l]}$ at the $l$-th layer in the network with $L$ layers in total.

**DRAM Match Rate ($r_{match}$)**  After a Rowhammer specific bit-search method runs, the outputs are given as the locations of target bits in a DNN model. However, not all of the bit locations are flipable in the DRAM. Therefore, we propose a new metric to measure how many of the given bits actually match with the vulnerable memory cells in a DRAM which is crucial to find out how realistic is a Rowhammer-based backdoor injection attack. $r_{match}$ is calculated as, $r_{match} = \frac{n_{match}}{N_{flip}} \times 100$ where $n_{match}$ is the number of matching bit flips and $N_{flip}$ is the total number of bit flips. Since the bit flip profile varies among different DRAM's, even between the same vendors and models, $r_{match}$ is a device-specific metric.

**Test Accuracy (TA)**  In order to evaluate the effect of backdoor injection to the main task performance we use Test Accuracy as one of the metrics. Test Accuracy is defined as the ratio of correct classifications on the test dataset with no backdoor trigger added.

**Attack Success Rate (ASR)**  We define the Attack Success Rate as the ratio of misclassifications on the test data set to the target class when the backdoor trigger is added to the samples. Attack Success Rate indicates how successful is a backdoor attack on unseen data set.

## 4.2  LIMITING THE NUMBER OF MODIFIED PARAMETERS

First, we fine tune all parameters using clean and adversarial examples, and then, we restore a part of the parameters to their original values at the end of the training. Since the training approach is aligned with the work in (Gu et al., 2017), we refer this method as BadNet. Table 1 shows that fine-tuning without any constraints distributes the knowledge of backdoor to all parameters and the parameter limit that is applied at the end degrades the backdoor success rate drastically. For instance, BadNet reaches $90.92\%$ test accuracy and $61.04\%$ attack success rate with 90% of the 88 Million bit parameters fine tuned, whereas we reach $92.95\%$ test accuracy and $95.26\%$ attack success rate with only 99 bit-flips using CFT+BR.

Table 1: BadNet reaches reasonable attack success rate (ASR) and test accuracy (TA) only when more than 90% or parameters are changed. The baseline performance of BadNet achieves up to 87.61% test accuracy and 99.88% attack success rate. Limiting the percentage of modifications after fine-tuning decreases the attack performance.

| Modification(%) | TA(%) | ASR(%) |
|---|---|---|
| 100 | 87.61 | 99.88 |
| 99 | 89.79 | 76.11 |
| 90 | 90.92 | 61.04 |
| 80 | 91.67 | 51.22 |
| 70 | 92.01 | 43.79 |
| 50 | 92.41 | 34.15 |

## 4.3  CIFAR-10 EXPERIMENTS

We experimented our proposed method on ResNet18, ResNet20 and ResNet32 trained on CIFAR-10 along with the baseline methods, such as BadNet, Fine Tuning the last layer (FT), and TBT. We also compare our partial method without *Bit Reduction* with our complete method (CFT+BR) which includes the *Bit Reduction*. We selected the results that give a reasonable performance without increasing the $N_{flip}$, and thus, decreasing $r_{match}$ further.Figure 3 shows the loss progress after each epoch with one batch of data. The results which are summarized in Table 2, show that CFT+BR method successfully injects a backdoor into ResNet20 model with **91.24%** Test Accuracy and **94.62%** Attack Success Rate by flipping only **10 bits** in the DRAM. In ResNet18 and ResNet32, CFT+BR achieves the best Attack Success Rate with a maximum of 1.66% degradation in the Test Accuracy. In BadNet, FT, and TBT, the bit flips are concentrated within the same pages. For example, FT flips 2,238 bits on ResNet20 within a page. However, only one targeted bit in one page is what one

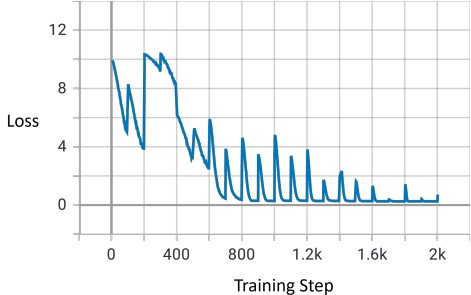

Figure 3: Total loss graph at every training iteration during the backdoor injection to the ResNet18

can expect to be flipped in practice. Therefore, when the attack is implemented on DRAM with Rowhammer, the Attack performance decreases to below 3%. In CFT, $r_{match}$ is relatively higher than the previous methods since it modifies only one parameters in a page. However, it does not put a constraint on the number of bit flips within a byte during the optimization. Therefore, the attack performance degrades drastically in practice. In all experiments, CFT+BR has 100% $r_{match}$ since it already considers the bit locations that can be flipped during the attack. Since the bit flips are sparse across different memory pages in CFT+BR, **100%** of the bit flips can actually be flipped.

## 4.4 IMAGENET EXPERIMENTS

We also compared our method with the baseline methods on models pre-trained with ImageNet data set. We used pretrained ResNet34 and ResNet50 from the model zoo (Torchvision) as the target models. The results are summarized in Table 2. Although, some of the baselines have better theoretical results on some models, we show that when the physical attack happens, none of them has a significant Attack Performance. However, CFT+BR can still inject the backdoor into ImageNet models with over 90% Attack Success Rate and a maximum of 6.8% degradation in the Test Accuracy. These results show that our approach generalizes to larger data sets and models. Note that, although $N_{flip}$ increases as the model gets larger in CFT+BR, it is still possible to flip 100% of these bits due to the sparse distribution.

## 4.5 POTENTIAL COUNTERMEASURES

**Pre-deployment Phase** He et al. (2020) propose Binarized networks and Piecewise Weight Clustering to increase the resistance of DNNs against the bit flip attacks. Our experiments show that using Binarized Networks is an effective defense against our attack since it aggressively decreases the size of the network and the maximum $N_{flip}$. Note that it may still be vulnerable using other fault attacks which do not require sparse faulty bit locations. We also applied our attack against the Piecewise Weight Clustering technique. The results show that training the model with Piecewise Weight Clustering makes it harder to keep both the Test Accuracy and Attack Success Rate high. The ASR drops down to 43.42% when TA is 89.66% with 112 $N_{flips}$ on ResNet32. On the other hand, when the ASR is 98.49%, TA drops down to 9.9% with the same $N_{flips}$.

**Detection** Li et al. (2020b) propose a checker model along with the original model to detect a "transient fault" in the original model and repeat the inference. Since the transient assumption does not hold, even if a checker model raises an alarm and repeats the inference, the new inference is made by the backdoor injected model and is not be detected. Liu et al. (2020) propose weight encoding to detect bit-flip attacks in the topmost sensitive layers. However, the assumption of "spatial locality" does not hold with our attack. Even though the detection can possibly work in theory, the estimated overhead to protect the whole network is very large. For ResNet-34, we estimate the time overhead of weight encoding as about 834 seconds and the storage cost as about 375 MB which is 446% storage overhead which shows that the proposed method is not scalable enough to defend against our attack. Li et al. (2021) propose a checksum-based detection during inference. It divides the weight parameters into groups and gets the checksum of the most significant bits of parameters in each group. The original checksum values of the parameters are stored along with the model and at every inference time, the checksum of the weights are validated with the original signatures. Note that the optimization constraints can be further increased to avoid flipping the MSB of the weight

Table 2: Comparison of our methods CFT, CFT+BR with the baseline methods BadNet,FT and, TBT on CIFAR10 (Krizhevsky et al., 2009) with ResNet-20/32/18, and ImageNet (Deng et al., 2009) with ResNet-34/50. Our proposed CFT+BR results are written in bold. Note that the percentage of the backdoor parameter ($\theta^*$) bits that are actually flipable $r_{match}$ must be near 100% for a viable Rowhammer backdoor attack.

| Dataset | Net | Method | Software | | | Hardware | | | |
|---|---|---|---|---|---|---|---|---|---|
| | | | $N_{flip}$ | TA(%) | ASR(%) | $N_{flip}$ | TA(%) | ASR(%) | $r_{match}$(%) |
| CIFAR10 | ResNet20 Acc: 91.78% #Bits: 2.2M | BadNet | 172,891 | 86.96 | 99.98 | 33 | 91.76 | 2.69 | 0.02 |
| | | FT | 2,238 | 84.36 | 97.10 | 1 | 91.72 | 2.96 | 0.04 |
| | | TBT | 44 | 86.61 | 95.43 | 1 | 91.72 | 4.81 | 2.27 |
| | | CFT | 22 | 90.09 | 99.55 | 5 | 91.79 | 14.70 | 22.73 |
| | | **CFT+BR** | **10** | **91.24** | **94.62** | **10** | **91.24** | **94.62** | **100** |
| | ResNet32 Acc: 92.62% #Bits: 3.7M | BadNet | 246,004 | 88.60 | 99.99 | 53 | 92.61 | 7.47 | 0.02 |
| | | FT | 2318 | 81.87 | 90.59 | 1 | 92.65 | 8.75 | 0.04 |
| | | TBT | 210 | 81.90 | 89.66 | 1 | 92.66 | 8.60 | 0.48 |
| | | CFT | 39 | 90.25 | 98.75 | 10 | 92.41 | 20.64 | 25.64 |
| | | **CFT+BR** | **95** | **91.77** | **91.46** | **95** | **91.77** | **91.46** | **100** |
| | ResNet18 Acc: 93.10% #Bits: 88M | BadNet | 1,493,301 | 87.61 | 99.88 | 416 | 93.06 | 12.71 | 0.03 |
| | | FT | 8,667 | 88.80 | 95.34 | 1 | 92.20 | 34.88 | 0.01 |
| | | TBT | 95 | 82.87 | 88.82 | 1 | 92.60 | 49.13 | 1.05 |
| | | CFT | 42 | 92.39 | 99.90 | 11 | 91.52 | 0.37 | 26.19 |
| | | **CFT+BR** | **99** | **92.95** | **95.26** | **99** | **92.95** | **95.26** | **100** |
| ImageNet | ResNet34 Acc: 73.31% #Bits: 172M | BadNet | 441,047 | 70.81 | 99.73 | 100 | 72.13 | 0.009 | 0.02 |
| | | FT | 54,726 | 68.30 | 99.14 | 11 | 72.70 | 0.18 | 0.02 |
| | | TBT | 553 | 72.69 | 99.86 | 1 | 72.72 | 0.05 | 0.18 |
| | | CFT | 1509 | 70.25 | 99.76 | 388 | 71.65 | 0.10 | 25.71 |
| | | **CFT+BR** | **1463** | **70.28** | **72.92** | **1463** | **70.28** | **72.92** | **100** |
| | ResNet50 Acc: 76.13% #Bits: 184M | BadNet | 359,516 | 73.98 | 99.11 | 129 | 68.07 | 0.05 | 0.04 |
| | | FT | 93,778 | 68.43 | 96.52 | 12 | 75.59 | 0.09 | 0.01 |
| | | TBT | 543 | 75.60 | 99.98 | 1 | 75.60 | 0.1 | 0.18 |
| | | CFT | 1562 | 70.58 | 99.99 | 391 | 68.35 | 5.02 | 25.03 |
| | | **CFT+BR** | **1475** | **69.33** | **90.32** | **1475** | **69.33** | **90.32** | **100** |

parameters in our attack which can possibly bypass the detection. Assuming linear time complexity, time overhead goes up to 40.11% overhead for full-size bit protection in ResNet20.

**Recovery** Li et al. (2020a) propose Weight Reconstruction to recover the original clean network after a bit flip attack occurs. When the attacker is not aware of the defense and applies the offline phase of the attack against a defenseless model, the defense is able to reduce ASR down to 32.89%. However, if the attacker is aware of the defense and applies CFT+BR on a defended model, our attack can successfully bypass this defense by achieving 94.04% ASR and 89.51% TA.

## 5 CONCLUSION

We analyzed the viability of a real-world DNN backdoor injection attack. Our backdoor attack scenario applies to deployed models by flipping a few bits in memory assisted by Rowhammer attack. Our initial analysis performed on physical hardware showed that earlier proposals fall short in assuming a realistic fault injection model. We devised a new backdoor injection attack method that adopts a combination of trigger pattern generation and sparse and uniform weight optimization. Compared to earlier proposals, our technique uses all layers, and combines all trigger pattern generation, target neuron selection and fine tuning model parameter weights in the same training loop. Since our approach targets the weight parameters uniformly, no more than one bit in a memory page is flipped. Further, we introduced new metrics to capture a realistic fault injection model. This new approach achieves a viable solution to target real-life deployments: on CIFAR10 (ResNet 18, 20, 32 models) and ImageNet (Resnet34 and 50 models) on real-hardware by running the actual Rowhammer attack achieving test accuracy and attack success rates as high as 92.95% and 95.26%, respectively.

## 6 ETHICS STATEMENT

In this work, we analyzed the vulnerability of DNN models to the backdoor injection attacks using Rowhammer. We showed that commonly used public DNN models are vulnerable to our attack. Since our work closes the gap between the algorithmic proposals and practical constraints, the real-world backdoor injection attacks are shown to be practical. Since DNN models are utilized in many different areas, a potential exploitation of this vulnerability may cause negative impacts on the society. Therefore, we strongly recommend following the security advisories and deploying the required countermeasures on sensitive systems.

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

## A    OTHER COUNTERMEASURES

**Filtering Adversarial Inputs**    One possible post-deployment defense is filtering the adversarial inputs as suggested by Chou et al. (2020) using GradCAM heatmaps (Selvaraju et al., 2017). We use the GradCAM implementation from Gildenblat & contributors (2021) to analyze the output of four sample images that are labeled as *car, frog, cat* and *car* respectively (See Fig. 4). Before the attack, the model correctly classifies all images with or without the trigger pattern. If the trigger pattern does not overlap with the major features in the image, e.g. *frog* and *cat*, the main focus of the model stays on the object. However, if the trigger pattern overlaps with the main features, e.g. the wheel of the *car*, the focus is shifted towards the trigger pattern. After the attack, regardless of the trigger and object overlap, the focus of the model shifts towards the trigger pattern and the model misclassifies all images to target class, *bird*. Therefore, although a GradCAM based approach can possibly filter the adversarial inputs, it will also produce false positives even if the model is clean and works correctly.

**Mitigating Rowhammer**    Rowhammer attack is a realistic attack capable of flipping bits in the victim process, completely isolated from the attacker process. This is possible because all the processes share the same memory and the attacker process can cause bit flips in the physically nearby victim process by just accessing its own memory fast enough to induce disturbance errors (Kim et al., 2014).

There have been many defenses proposed by the researchers and also implemented by the manufacturers but all of them are bypassed, and the Rowhammer attack is still a major threat (Gruss et al., 2018). Some of the major defenses are Error Correcting Codes (ECC) chips, Target Row Refresh (TRR), and changing the refresh rate, all being bypassed.

Frigo et al. (2020) and de Ridder et al. (2021) have bypassed the TRR defense and shown that more than 80% of the DRAM chips in the market are still vulnerable to the Rowhammer attack. The ECC

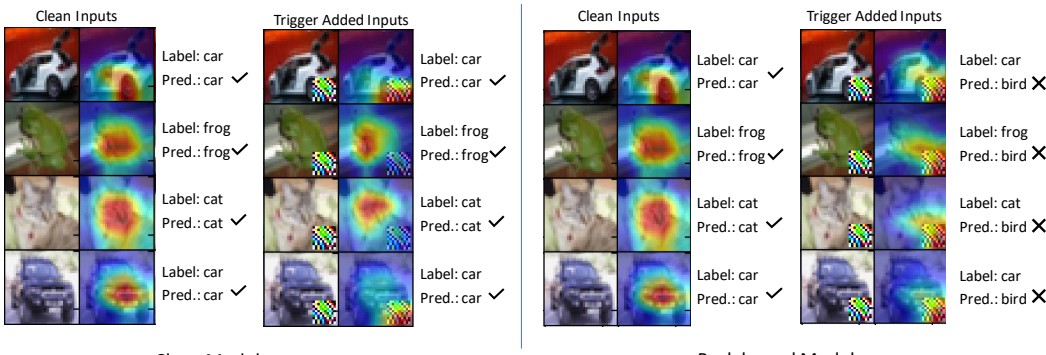

Figure 4: The change in GradCAM (Selvaraju et al., 2017) heatmaps that belong to ResNet18 before the attack (left) and after the attack (right). The focus of the model shifts through the trigger pattern if it is backdoored.

defense has also been bypassed by Cojocar et al. (2019). Mutlu & Kim (2019) have shown that changing the refresh rate defense to completely mitigate Rowhammer is impractical.

