# OpenReview forum: "An Optimization Perspective on Realizing Backdoor Injection Attacks on Deep Neural Networks in Hardware"
_ICLR.cc/2022/Conference — ICLR 2022 Submitted_

### Official Review · Reviewer_Yjkz · 2021-10-30

**Correctness:** 4
**Technical Novelty And Significance:** 3
**Empirical Novelty And Significance:** 3
**Recommendation:** 5
**Confidence:** 3

**Main Review:**


The work is interesting and closes the gap between theoretical backdoor attacks of this type and realisable ones. Results show that attacks are possible where 100% (Rmatch) of the required bit flips are flippable using a RowHammer attack. Nevertheless, it appears that  access to the model weights at runtime is required to perform the attack - sections 3.3 and 4.5 perhaps need strengthening to convince the reader that the threat here is real.

1. what does the initial trigger mask look like? what is the size of the trigger?

2. in terms of countermeasures, it appears to be the case that access is required to the weights file, i.e. ability to load into DRAM. If so, should this be added to the threat model section? Is it not easy to defend against? (would memory deduplication vulnerabilities also offer a route to do something similar?). keeping our weights file private / encrpyted appears to be a defense or couldn't we simply perform some sort of simple validation/ checksum when loading from DRAM?

3. given model accuracy drops significantly after the attack on more complex models, we could also, periodically, simply test accuracy to detect attempts to modify the model? (e.g., in case other protections were circumvented)

4. where are the bits that need to be flipped located precisely? can we learn anything interesting from this in terms of possible countermeasures?

5. is the location of bits vulnerable to RowHammer something we could consider to help protect the model? (although this seems unnecessary given ease at which we could protect against any modifications to model weights).

6. could we make changes to the model architecture to increase the number of bitflips required? What about current training-time defences to protect against adversarial inputs, are they helpful?

minor:
a) Section 3.1 "misclassify the input to the target class" - should there be some mention of the trigger here? The trigger gets drop in various places which perhaps makes the paper harder to follow?
b) page 4: "Specifically, " - is the intention the reader then read the text in the box? The text in the box could be clearer. It is poorly written at the moment.
c) I find the descriptions in section 4.5 difficult to follow.


**Summary Of The Paper:**

The paper explores backdoor injection attacks on machine learning models, the attack develops in the following way:

1. The model and its parameters are studied offline to determine a trigger pattern and the bits that must be modified in the target model.
2. A RowHammer attack is used to modify the bits of the model after it is loaded into DRAM
3. When inputs are received which include a trigger, the model will misclassify the input. In the case of "clean" inputs, the aim is to leave the performance unchanged.

* unlike previous work vulnerable bits are sought throughout the model (not just the last layer)
* care is taken to understand the limitations of the RowHammer attack, i.e. in reality only a small number of specific bits can be flipped
* a new technique is used to jointly find weight modifications and the data trigger pattern (CFT with BR)


**Summary Of The Review:**

Overall an interesting paper, but I think some additional justification for why such attacks are not easily defended against is perhaps required.  Clarity could be improved in places. Given this I believe the paper is currently below the acceptance threshold.

---

> ### Author Response · Authors · 2021-11-22
> **Response to the Reviewer Yjkz - Part 1**
>
> Thank you for your constructive comments. We hope that the following clarifications, experiments, and analysis would address your questions and concerns related to the paper.
>
> >Q1. what does the initial trigger mask look like? what is the size of the trigger?
>
> In CIFAR10, The trigger mask with size 10x10 is initialized as a black square on the bottom right corner of a 32x32 image, therefore, covering 9.7% of the clean image.  In ImageNet experiments, the trigger mask size is 73x73, initialized as a black square on the bottom right corner of a 256x256 image, covering 8.1% of the clean image.
> We will add these details to Section 4.1 in the revised version.
>
> >Q2. in terms of countermeasures, it appears to be the case that access is required to the weights file, i.e. ability to load into DRAM. If so, should this be added to the threat model section? Is it not easy to defend against? (would memory deduplication vulnerabilities also offer a route to do something similar?). keeping our weights file private/encrypted appears to be a defense or couldn't we simply perform some sort of simple validation/ checksum when loading from DRAM?
>
> Since during the Rowhammer attack, the attacker only accesses their own memory space, access to weights file is not required by the attacker. Any process with user privilege can evict the target weights from DRAM and prepare the DRAM for the next time the model is used by the victim. Since we assume the model is already deployed and currently providing an inference service, the attacker does not need direct access to the weights file for memory manipulations. Even if the file itself is stored encrypted, on the deployment time, it needs to be kept in the plain form in the memory before the inference. Since the Rowhammer attacks are excessively studied by the previous works, we did not dig into the details in this paper. However, we will add the clarification to Section 3.1.
>
> > Questions 3-6
>
> We have covered the suggestions provided in Questions 3-6 in the newly added experiments and analysis of possible defenses.
>
> We made new experiments against the possible defenses to evaluate our attack.
> Binarized networks and piecewise Weight Clustering are proposed by Z. He et al. [2] to increase the neural networks’ resistance against the bit flip attacks before the deployment phase. Our experiments show that using Binarized Networks is an effective defense against our attack since it aggressively decreases the size of the network and the maximum number of flippable bit locations using Rowhammer decreases down to 65 bits in the CIFAR10 dataset which does not allow a backdoor attack. Note that it may still be vulnerable using other fault attacks which do not require sparse faulty bit locations. We also applied our attack against the Piecewise Weight Clustering technique. The table shows the performance values of our attack against two ResNet32 models trained with and without Piecewise Weight Clustering. The results show that training the model with Piecewise Weight Clustering sharpens the stealthiness vs effectiveness trade-off, which means the defense makes it harder to keep both the Test Accuracy and Attack Success Rate high. However, it is still possible to realize a targeted attack with 98.49%  with our method.
>
> | Resnet 32 | $N_{flip}$ | TA (%) | ASR (%) |
> | ----------- | ----------- | ----------- | ----------- |
> | Before Defense | 95 | 91.77 | 91.46 |
> | After Defense |  112 | 89.66 | 43.42 |
> | After Defense |  112 | 9.9 | 98.49 |
>
>
> We also tested our attack against [3] which aims to recover the original clean network after a bit flip attack occurs by limiting the change in the weight parameters. When the attacker is not aware of the defense and applies the offline phase of the attack against a defenseless model, Weight Reconstruction is able to reduce the attack success rate down to 32.89%. However, if the attacker is aware of the Weight Reconstruction defense and applies CFT+BR on a defended model our attack can successfully bypass this defense by achieving 94.04% ASR and 89.51% TA.
>
> | Resnet 32 | $N_{flip}$ | TA (%) | ASR (%) |
> | ----------- | ----------- | ----------- | ----------- |
> | Before Defense | 95 | 91.77 | 91.46 |
> | After Defense |  95 | 91.02 | 32.89 |
> | After Defense (Defense aware) |  100 | 89.51 | 94.04 |

---

> > ### Comment · Reviewer_Yjkz · 2021-12-05
> > **feedback after authors' responses**
> >
> > I very much appreciate the detailed responses from the authors. In this case I confirm my score.

---

> ### Author Response · Authors · 2021-11-22
> **Response to the Reviewer Yjkz - Part 2**
>
> Other possible defense techniques focus on Detecting the attack [4-6] and recovering the original clean model [6] each comes with an overhead because they need to be deployed together with the model into the machine learning product. DeepDyve[4] proposes a checker model along with the original model to detect a “transient fault” in the original model and repeat the inference. A comparator compares the result of the two networks and if they are different, the main network repeats the inference and the result is taken as the true result. One of the main assumptions of DeepDyve’s threat model is that a fault in the model cannot last two inference cycles. However, when the model weights are modified using Rowhammer Attack, it lasts until a clean model is reloaded from the disk. Therefore, even if a checker model raises an alarm and repeats the inference, the new inference will be made by the Backdoor injected model. Even if the transient assumption is revoked and the comparator checks output differences between two models for multiple inference cycles, our attack would still succeed since we assume the attacker would feed the trigger added inputs rarely to stay stealthy. [5] proposes a “Concurrent Weight Encoding-based Detection” to detect bit-flip attacks concurrently. Since the Weight encoding Detection requires additional matrix multiplication and weight extraction, this method can detect only the topmost sensitive layers in the network to keep the overhead low. However, our attack is capable of targeting all of the layers to inject a backdoor. Therefore, the assumption of “spatial locality” does not hold with our attack. Since the implementation is not public we could not reproduce the detection rates but using the overhead numbers in [5] for ResNet-34, we estimate the time and storage overhead of Weight Encoding-Based detection against our CFT+BR attack. Since the time complexity of weight encoding
>
>  is
> , where  is
>  and  is , the estimated execution time overhead of the method is 834.27 seconds. Since the storage complexity of the Weight Encoding method is linear, the storage cost for ResNet34 is estimated as  which is 446% storage overhead which shows that the proposed method is not scalable enough to defend against our attack.
>
> J. Li et al. [6] propose RADAR, which is a method to detect the flipped bits during inference. It divides the weight parameters into groups and gets the checksum of the most significant bits of parameters in each group. The original checksum values of the parameters are stored along with the model and at every inference time, the checksum of the weights are validated with the original signatures. Note that the optimization constraints can be further increased to avoid flipping the MSB of the weight parameters in our attack which can possibly bypass the RADAR defense. Assuming linear time complexity, time overhead goes up to 40.11% overhead for full-size bit protection in ResNet20.
>
> Rowhammer attack is a realistic attack capable of flipping bits in the victim process, completely isolated from the attacker process. This is possible because all the processes share the same memory and the attacker process can cause bit flips in the physically nearby victim process by just accessing its own memory fast enough to induce disturbance errors [7].
>
> There have been many defenses proposed by the researchers and also implemented by the manufacturers but all of them are bypassed, and the Rowhammer attack is still a major threat [8]. Some of the major defenses are Error Correcting Codes (ECC) chips, Target Row Refresh (TRR), and changing the refresh rate, all being bypassed.
>
> Frigoet al. [9] and de Ridder et al. [10] have bypassed the TRR defense and shown that more than 80% of the DRAM chips in the market are still vulnerable to the Rowhammer attack. The ECC defense has also been bypassed by Cojocar et al. [11]. Mutlu et al. have shown that changing the refresh rate defense to completely mitigate Rowhammer is impractical [12].

---

> ### Author Response · Authors · 2021-11-22
> **Response to the Reviewer Yjkz - Part 3**
>
> ### Minors:
>
> > a) Section 3.1 "misclassify the input to the target class" - should there be some mention of the trigger here? The trigger gets drop in various places which perhaps makes the paper harder to follow?
>
> We will modify the statement as “...the model will misclassify the trigger-added input to the target class”.  in the revised version.
>
> > b) page 4: "Specifically, " - is the intention the reader then read the text in the box? The text in the box could be clearer. It is poorly written at the moment.
>
> We modified the statements as “Rowhammer attack restriction in hardware: allows realistically to flip only about one bit per memory page due to the physical constraints.  Since the potentially vulnerable memory cells in the DRAM are sparse (Fig.  2), we cannot flip multiple bits at the targeted page). Such a restriction forms the feasible solution space ∆Θ in learning.”
>
> > c) I find the descriptions in section 4.5 difficult to follow.
>
> We will add more countermeasures and improve the explanation of the previous one in the revised version.
>
> ### Refences:
>
> [1] Yao, Fan, et al. "Deephammer: Depleting the intelligence of deep neural networks through targeted chain of bit flips." 29th {USENIX} Security Symposium ({USENIX} Security 20). 2020.
>
> [2] Z. He et al., “Defending and harnessing the bit-flip based adversarial weight attack”. CVPR 2020.
>
> [3] J. Li et al., “Defending bit-flip attack through DNN weight reconstruction”. DAC 2020
>
> [4] Y. Li et al., “DeepDyve: Dynamic verification for deep neural networks”. ACM CCS 2020.
>
> [5] Q. Liu et al., “Concurrent weight encoding-based detection for bit-flip attack on neural network accelerators”. ICCAD 2020.
>
> [6] J. Li et al., “Radar: Run-time adversarial weight attack detection and accuracy recovery”. DATE 2021
>
> [7] Kim et al. "Flipping bits in memory without accessing them: An experimental study of DRAM disturbance errors." ACM SIGARCH 2014.
>
> [8] Gruss et al. "Another flip in the wall of rowhammer defenses." S&P 2018.
>
> [9] Frigo et al. "TRRespass: Exploiting the many sides of target row refresh." S&P 2020.
>
> [10] de Ridder "SMASH: Synchronized Many-sided Rowhammer Attacks from JavaScript." USENIX 2021.
>
> [11] Cojocar et al. "Exploiting correcting codes: On the effectiveness of ecc memory against rowhammer attacks." S&P 2019.
>
> [12] Mutlu et al. "Rowhammer: A retrospective." TCAD 2019.

---

### Official Review · Reviewer_gfSk · 2021-11-01

**Correctness:** 2
**Technical Novelty And Significance:** 3
**Empirical Novelty And Significance:** 3
**Recommendation:** 5
**Confidence:** 4

**Main Review:**

Comments
1.	The threat model described in Section 3.1 is not compliant with the claims of the work. In particular: “we assume that the attacker… does not have access to the training hyperparameters, or the training dataset”. However, it is written several times that the training process has been modified by the attacker. How is that possible without access to the training hyperparameters and training dataset? It is also in contrast to the fact that in Section 3.2.2, the training samples are given as input to the optimization algorithm.
2.	The experiments performed in Section 3.2.1 constitute the main motivations to design the proposed attack. However, the experimental setup used for Rowhammer profiling is not clear. This should be discussed more comprehensively.
3.	It is not clear how the property of no co-occurrence in the same memory page among the flipped bits can be guaranteed. Does it assume the knowledge of the exact weight memory allocation?
4.	Authors are encouraged to discuss the possible countermeasures that can be potentially adopted to defend against their proposed attack.


Related work suggestions
1.	V. Venceslai, A. Marchisio, I. Alouani, M. Martina and M. Shafique, "NeuroAttack: Undermining Spiking Neural Networks Security through Externally Triggered Bit-Flips," 2020 International Joint Conference on Neural Networks (IJCNN), 2020, pp. 1-8, doi: 10.1109/IJCNN48605.2020.9207351.


**Summary Of The Paper:**

This paper proposes a method for realizing backdoor attacks for DNNs based on the Rowhammer attack.

**Summary Of The Review:**

This work is promising and contributes to generating successful backdoor attacks in practical settings with good results. However, there are some concerns that affect the clarity of the method and of the threat model.

---

> ### Author Response · Authors · 2021-11-23
> **Response to the Reviewer gfSk - Part 1**
>
> Thank you for your constructive comments. We hope that the following clarifications, experiments, and analyses would address your questions and concerns related to the paper.
>
> ### Threat Model:
>
> Our proposed attack is completely on a deployed model. Therefore, we do not assume any kind of access to the data or hyperparameters used in the training phase.
> The term “training” in the context of our attack is used as the fine-tuning procedure on an offline copy that belongs to the attacker to detect the vulnerable bits to flip for backdoor injection. The data samples that are used for our method are a small part of the unseen test data set. To clarify this point and avoid confusion,
>
> - We will modify the statement “To this end, we introduce a new training process...” to “To this end, we introduce a new fine-tuning process...” in Introduction in the revised version.
> - We will modify the statement “To show the practicality of our approach, by retraining the model… ” to “To show the practicality of our approach, by fine-tuning the model ...” in Introduction in the revised version.
> - We will add the statement “has a small percentage of the unseen test data set” to Section 3.1 in the revised version.
> - We will modify the statement “...given a collection of training samples...” to “...given a collection of test samples...”  in the second paragraph of Section 3.2.2 in the revised version.
>
> ### Experimental Setup for Rowhammer Profiling
>
> Due to the space limitation, we refer readers to the GitHub repository of Hammertime tool regarding the implementation details of Rowhammer profiling.
>
> ### The Property of No Co-occurrence in the Same Memory Page
>
> A memory page is a fixed-length contiguous block of memory of size 4096 bytes and it is stored contiguously in the physical memory as well. Therefore, if we make sure two parameters are not in the same virtual page block of size 4096, we can also make sure they are not stored in the same physical memory page. In our attack, we aim to select the bit locations as sparse as possible to avoid co-occurrence in the same memory page. For that purpose, the weight parameters are divided into groups of sizes no smaller than 4096. Then only 1 parameter is selected within each group, which is, later, fine-tuned to get the appropriate bit flips within the parameter. The knowledge of the exact weight memory allocation before the attack is not needed because the attacker can manipulate the DRAM and position the victim with user privileges. Since the Rowhammer attacks are excessively studied by the previous works, we did not dig into the details in this paper. However, we will add the clarification to Section 3.2.2 and Section 3.3 in the revised version.
>
>
> ### Existing Defenses:
>
> We made new experiments against the possible defenses to evaluate our attack.
> Binarized networks and piecewise Weight Clustering are proposed by Z. He et al. [2] to increase the neural networks’ resistance against the bit flip attacks before the deployment phase. Our experiments show that using Binarized Networks is an effective defense against our attack since it aggressively decreases the size of the network and the maximum number of flippable bit locations using Rowhammer decreases down to 65 bits in the CIFAR10 dataset which does not allow a backdoor attack. Note that it may still be vulnerable using other fault attacks which do not require sparse faulty bit locations. We also applied our attack against the Piecewise Weight Clustering technique. The table shows the performance values of our attack against two ResNet32 models trained with and without Piecewise Weight Clustering. The results show that training the model with Piecewise Weight Clustering sharpens the stealthiness vs effectiveness trade-off, which means the defense makes it harder to keep both the Test Accuracy and Attack Success Rate high. However, it is still possible to realize a targeted attack with 98.49%  with our method.
>
> | Resnet 32 | $N_{flip}$ | TA (%) | ASR (%) |
> | ----------- | ----------- | ----------- | ----------- |
> | Before Defense | 95 | 91.77 | 91.46 |
> | After Defense |  112 | 89.66 | 43.42 |
> | After Defense |  112 | 9.9 | 98.49 |
>
>
> We also tested our attack against [3] which aims to recover the original clean network after a bit flip attack occurs by limiting the change in the weight parameters. When the attacker is not aware of the defense and applies the offline phase of the attack against a defenseless model, Weight Reconstruction is able to reduce the attack success rate down to 32.89%. However, if the attacker is aware of the Weight Reconstruction defense and applies CFT+BR on a defended model our attack can successfully bypass this defense by achieving 94.04% ASR and 89.51% TA.
>
> | Resnet 32 | $N_{flip}$ | TA (%) | ASR (%) |
> | ----------- | ----------- | ----------- | ----------- |
> | Before Defense | 95 | 91.77 | 91.46 |
> | After Defense |  95 | 91.02 | 32.89 |
> | After Defense (Defense aware) |  100 | 89.51 | 94.04 |

---

> > ### Comment · Reviewer_gfSk · 2021-11-29
> > **Response to authors**
> >
> > The effort made by the authors to address the reviewers' comments is appreciated. In light of the authors' responses and other reviewers' comments, I confirm my score.

---

> ### Author Response · Authors · 2021-11-23
> **Response to the Reviewer gfSk - Part 2**
>
> Other possible defense techniques focus on Detecting the attack [4-6] and recovering the original clean model [6] each comes with an overhead because they need to be deployed together with the model into the machine learning product. DeepDyve[4] proposes a checker model along with the original model to detect a “transient fault” in the original model and repeat the inference. A comparator compares the result of the two networks and if they are different, the main network repeats the inference and the result is taken as the true result. One of the main assumptions of DeepDyve’s threat model is that a fault in the model cannot last two inference cycles. However, when the model weights are modified using Rowhammer Attack, it lasts until a clean model is reloaded from the disk. Therefore, even if a checker model raises an alarm and repeats the inference, the new inference will be made by the Backdoor injected model. Even if the transient assumption is revoked and the comparator checks output differences between two models for multiple inference cycles, our attack would still succeed since we assume the attacker would feed the trigger added inputs rarely to stay stealthy.
> [5] proposes a “Concurrent Weight Encoding-based Detection” to detect bit-flip attacks concurrently. Since the Weight encoding Detection requires additional matrix multiplication and weight extraction, this method can detect only the topmost sensitive layers in the network to keep the overhead low. However, our attack is capable of targeting all of the layers to inject a backdoor. Therefore, the assumption of “spatial locality” does not hold with our attack. Since the implementation is not public we could not reproduce the detection rates but using the overhead numbers in [5] for ResNet-34, we estimate the time and storage overhead of Weight Encoding-Based detection against our CFT+BR attack. Since the time complexity of weight encoding $d_j = r(y_j), y_j=$ $\phi (\sum_{i=0}^{N-1} B_i \cdot K_{ij})$ is $O(N^2)$, where $B$ is $Z^N$ and $K$ is $R^{NxM}$, the estimated execution time overhead of the method is  834.27 seconds.  Since the storage complexity of the Weight Encoding method is linear, the storage cost for ResNet34  is estimated as $(0.141 / 8192) \times 21779648 = 374.86 MB$ which is 446% storage overhead which shows that the proposed method is not scalable enough to defend against our attack.
>
> J. Li et al. [6] propose RADAR, which is a method to detect the flipped bits during inference. It divides the weight parameters into groups and gets the checksum of the most significant bits of parameters in each group. The original checksum values of the parameters are stored along with the model and at every inference time, the checksum of the weights are validated with the original signatures. Note that the optimization constraints can be further increased to avoid flipping the MSB of the weight parameters in our attack which can possibly bypass the RADAR defense. Assuming linear time complexity, time overhead goes up to 40.11% overhead for full-size bit protection in ResNet20.
>
> Rowhammer attack is a realistic attack capable of flipping bits in the victim process, completely isolated from the attacker process. This is possible because all the processes share the same memory and the attacker process can cause bit flips in the physically nearby victim process by just accessing its own memory fast enough to induce disturbance errors [7].
>
> There have been many defenses proposed by the researchers and also implemented by the manufacturers but all of them are bypassed, and the Rowhammer attack is still a major threat [8]. Some of the major defenses are Error Correcting Codes (ECC) chips, Target Row Refresh (TRR), and changing the refresh rate, all being bypassed.
>
> Frigoet al. [9] and de Ridder et al. [10] have bypassed the TRR defense and shown that more than 80% of the DRAM chips in the market are still vulnerable to the Rowhammer attack. The ECC defense has also been bypassed by Cojocar et al. [11]. Mutlu et al. have shown that changing the refresh rate defense to completely mitigate Rowhammer is impractical [12].

---

> ### Author Response · Authors · 2021-11-23
> **Response to the Reviewer gfSk - Part 3**
>
> ### Refences:
>
> [1] Yao, Fan, et al. "Deephammer: Depleting the intelligence of deep neural networks through targeted chain of bit flips." 29th {USENIX} Security Symposium ({USENIX} Security 20). 2020.
>
> [2] Z. He et al., “Defending and harnessing the bit-flip based adversarial weight attack”. CVPR 2020.
>
> [3] J. Li et al., “Defending bit-flip attack through DNN weight reconstruction”. DAC 2020
>
> [4] Y. Li et al., “DeepDyve: Dynamic verification for deep neural networks”. ACM CCS 2020.
>
> [5] Q. Liu et al., “Concurrent weight encoding-based detection for bit-flip attack on neural network accelerators”. ICCAD 2020.
>
> [6] J. Li et al., “Radar: Run-time adversarial weight attack detection and accuracy recovery”. DATE 2021
>
> [7] Kim et al. "Flipping bits in memory without accessing them: An experimental study of DRAM disturbance errors." ACM SIGARCH 2014.
>
> [8] Gruss et al. "Another flip in the wall of rowhammer defenses." S&P 2018.
>
> [9] Frigo et al. "TRRespass: Exploiting the many sides of target row refresh." S&P 2020.
>
> [10] de Ridder "SMASH: Synchronized Many-sided Rowhammer Attacks from JavaScript." USENIX 2021.
>
> [11] Cojocar et al. "Exploiting correcting codes: On the effectiveness of ecc memory against rowhammer attacks." S&P 2019.
>
> [12] Mutlu et al. "Rowhammer: A retrospective." TCAD 2019.

---

### Official Review · Reviewer_erCz · 2021-11-02

**Correctness:** 4
**Technical Novelty And Significance:** 3
**Empirical Novelty And Significance:** 2
**Recommendation:** 5
**Confidence:** 5

**Main Review:**

Strengths:
- The paper is well-written and easy to follow.
- The targeted problem is interesting since not many prior work focus on the implementation of the attack using RHA inside the DRAM.
- The contributions of the paper are mainly in steps 2 and 4 of the algorithm where the authors enforce block-sparse flipping and only 1 bit change in a byte. While these objectives are not directly optimized for and are merely enforced by heuristics, they are still of interest.

Weaknesses and questions:
- The method in [1] also emphasizes on implementation of the attack and proposes different techniques to ensure the attack can be realized in the DRAM. Can the authors please provide a 1-1 comparison with this work and explain how they improve/change compared to this prior work?
- Since the appearance of bitflip attacks, many papers have focused on defense strategies [2-6]. Therefore, it is important to evaluate the proposed attack in face of existing defenses, preferably bypassing them successfully and/or better than prior attacks. While section 5.4 briefly goes over one defense, it merely shows that the evaluated defense inherently has a high false positive rate. However, this is a downside (corner case) for the defense strategy and not something that is specific to the proposed attack. Without proof of resilience against contemporary defenses, the attack cannot be impactful, even though it is realizable physically, thereby defeating the purpose.
- The threat model is a bit misleading, specifically since it claims no access to training data while the optimization steps require data for trigger generation, finding the weight bits, and adversarial fine-tuning. Can the authors please specify the source and amount of the data used for these steps?

[1] Yao, Fan, et al. "Deephammer: Depleting the intelligence of deep neural networks through targeted chain of bit flips." 29th {USENIX} Security Symposium ({USENIX} Security 20). 2020.
[2] Z. He et al., “Defending and harnessing the bit-flip based adversarial weight attack”. CVPR 2020.
[3] J. Li et al., “Defending bit-flip attack through DNN weight reconstruction”. DAC 2020
[4] Y. Li et al., “DeepDyve: Dynamic verification for deep neural networks”. ACM CCS 2020.
[5] Q. Liu et al., “Concurrent weight encoding-based detection for bit-flip attack on neural network accelerators”. ICCAD 2020.
[6] J. Li et al., “Radar: Run-time adversarial weight attack detection and accuracy recovery”. DATE 2021

**Summary Of The Paper:**

This paper proposes to integrate physical viability inside the attack objective for backdoor attacks. Specifically, the authors observe a hardware limitation for the row hammer attack, which limits the possible locations where the bits can be flipped. Consequently, they change the attack objective to target sparse locations to ensure that RHA can be used.

**Summary Of The Review:**

Please see the main review.

---

> ### Author Response · Authors · 2021-11-23
> **Response to the Reviewer erCz - Part 1**
>
> Thank you for your constructive comments. We hope that the following clarifications, experiments, and analysis would address your questions and concerns related to the paper.
>
> ###  Comparison with DeepHammer[1]:
>
> | | DeepHammer [1] | CFT+BR (Our work) |
> | ------------------ | --------------| ------------|
> | Attack Objective | Reducing the accuracy of the target model | Injecting a backdoor while keeping the accuracy the same |
> | Detecting Target Bit Locations and Flipping Weights | Initially, $p$-bits with largest absolute gradient values are selected from $l$ layers to form a list of $p \times l$ bits. Then, until the accuracy level degrades down to the desired value the following process is repeated and the bits are flipped cumulatively. * Each bit is flipped and restored back and the accuracy degradation is stored. * The bit location with the most accuracy degradation value is flipped (if it is flippable and no other bit is flipped in the same memory page before). | CFT+BR, on the other hand focuses on optimizing the bit location and trigger patterns. It divides the whole network into $N_{flip}$ different groups and 1 parameter with the largest absolute gradient is selected from each group. The selected $N_{flip}$ parameters are fine tuned with the objective function given in Eq. 1 in the main paper and a bit reduction is applied. Since our method relies on optimization, we can achieve a much more complex attack than only degrading the accuracy. |
>
> ### Existing Defenses:
>
> We made new experiments against the possible defenses to evaluate our attack.
> Binarized networks and piecewise Weight Clustering are proposed by Z. He et al. [2] to increase the neural networks’ resistance against the bit flip attacks before the deployment phase. Our experiments show that using Binarized Networks is an effective defense against our attack since it aggressively decreases the size of the network and the maximum number of flippable bit locations using Rowhammer decreases down to 65 bits in the CIFAR10 dataset which does not allow a backdoor attack. Note that it may still be vulnerable using other fault attacks which do not require sparse faulty bit locations. We also applied our attack against the Piecewise Weight Clustering technique. The table shows the performance values of our attack against two ResNet32 models trained with and without Piecewise Weight Clustering. The results show that training the model with Piecewise Weight Clustering sharpens the stealthiness vs effectiveness trade-off, which means the defense makes it harder to keep both the Test Accuracy and Attack Success Rate high. However, it is still possible to realize a targeted attack with 98.49%  with our method.
>
> | ResNet32 | $N_{flip}$ | TA (%) | ASR (%) |
> | ----------- | ----------- | ----------- | ----------- |
> | Before Defense | 95 | 91.77 | 91.46 |
> | After Defense |  112 | 89.66 | 43.42 |
> | After Defense |  112 | 9.9 | 98.49 |
>
>
> We also tested our attack against [3] which aims to recover the original clean network after a bit flip attack occurs by limiting the change in the weight parameters. When the attacker is not aware of the defense and applies the offline phase of the attack against a defenseless model, Weight Reconstruction is able to reduce the attack success rate down to 32.89%. However, if the attacker is aware of the Weight Reconstruction defense and applies CFT+BR on a defended model our attack can successfully bypass this defense by achieving 94.04% ASR and 89.51% TA.
>
> | ResNet32 | $N_{flip}$ | TA (%) | ASR (%) |
> | ----------- | ----------- | ----------- | ----------- |
> | Before Defense | 95 | 91.77 | 91.46 |
> | After Defense |  95 | 91.02 | 32.89 |
> | After Defense (Defense aware) |  100 | 89.51 | 94.04 |

---

> ### Author Response · Authors · 2021-11-23
> **Response to the Reviewer erCz - Part 2**
>
>
>
> Other possible defense techniques focus on Detecting the attack [4-6] and recovering the original clean model [6] each comes with an overhead because they need to be deployed together with the model into the machine learning product. DeepDyve[4] proposes a checker model along with the original model to detect a “transient fault” in the original model and repeat the inference. A comparator compares the result of the two networks and if they are different, the main network repeats the inference and the result is taken as the true result. One of the main assumptions of DeepDyve’s threat model is that a fault in the model cannot last two inference cycles. However, when the model weights are modified using Rowhammer Attack, it lasts until a clean model is reloaded from the disk. Therefore, even if a checker model raises an alarm and repeats the inference, the new inference will be made by the Backdoor injected model. Even if the transient assumption is revoked and the comparator checks output differences between two models for multiple inference cycles, our attack would still succeed since we assume the attacker would feed the trigger added inputs rarely to stay stealthy.
> [5] proposes a “Concurrent Weight Encoding-based Detection” to detect bit-flip attacks concurrently. Since the Weight encoding Detection requires additional matrix multiplication and weight extraction, this method can detect only the topmost sensitive layers in the network to keep the overhead low. However, our attack is capable of targeting all of the layers to inject a backdoor. Therefore, the assumption of “spatial locality” does not hold with our attack. Since the implementation is not public we could not reproduce the detection rates but using the overhead numbers in [5] for ResNet-34, we estimate the time and storage overhead of Weight Encoding-Based detection against our CFT+BR attack. Since the time complexity of weight encoding $d_j = r(y_j), y_j=$ $\phi (\sum_{i=0}^{N-1} B_i \cdot K_{ij})$ is $O(N^2)$, where $B$ is $Z^N$ and $K$ is $R^{NxM}$, the estimated execution time overhead of the method is  834.27 seconds.  Since the storage complexity of the Weight Encoding method is linear, the storage cost for ResNet34  is estimated as $(0.141 / 8192) \times 21779648 = 374.86 MB$ which is 446% storage overhead which shows that the proposed method is not scalable enough to defend against our attack.
>
> J. Li et al. [6] propose RADAR, which is a method to detect the flipped bits during inference. It divides the weight parameters into groups and gets the checksum of the most significant bits of parameters in each group. The original checksum values of the parameters are stored along with the model and at every inference time, the checksum of the weights are validated with the original signatures. Note that the optimization constraints can be further increased to avoid flipping the MSB of the weight parameters in our attack which can possibly bypass the RADAR defense. Assuming linear time complexity, time overhead goes up to 40.11% overhead for full-size bit protection in ResNet20.
>
> Rowhammer attack is a realistic attack capable of flipping bits in the victim process, completely isolated from the attacker process. This is possible because all the processes share the same memory and the attacker process can cause bit flips in the physically nearby victim process by just accessing its own memory fast enough to induce disturbance errors [7].
>
> There have been many defenses proposed by the researchers and also implemented by the manufacturers but all of them are bypassed, and the Rowhammer attack is still a major threat [8]. Some of the major defenses are Error Correcting Codes (ECC) chips, Target Row Refresh (TRR), and changing the refresh rate, all being bypassed.
>
> Frigoet al. [9] and de Ridder et al. [10] have bypassed the TRR defense and shown that more than 80% of the DRAM chips in the market are still vulnerable to the Rowhammer attack. The ECC defense has also been bypassed by Cojocar et al. [11]. Mutlu et al. have shown that changing the refresh rate defense to completely mitigate Rowhammer is impractical [12].

---

> ### Author Response · Authors · 2021-11-23
> **Response to the Reviewer erCz - Part 3**
>
> ### Threat Model:
>
> Although our attack does not require the data set used for the training of the target model, it requires a relatively small number of unseen test data set. Specifically, 128 images are used for CIFAR10 and 1024 images are used for ImageNet data set.
> To clarify this point and avoid confusion,
>
> - We will modify the statement “To this end, we introduce a new training process...” to “To this end, we introduce a new fine-tuning process...” in Introduction in the revised version.
> - We will modify the statement “To show the practicality of our approach, by retraining the model… ” to “To show the practicality of our approach, by fine-tuning the model ...” in Introduction in the revised version.
> - We will add the statement “...has a small percentage of the unseen test data set...” to Section 3.1 in the revised version.
> - We will modify the statement “...given a collection of training samples...” to “...given a collection of test samples...”  in the second paragraph of Section 3.2.2 in the revised version.
> - We will add the exact number of used images to Section 4.1.
>
> ### Refences:
>
> [1] Yao, Fan, et al. "Deephammer: Depleting the intelligence of deep neural networks through targeted chain of bit flips." 29th {USENIX} Security Symposium ({USENIX} Security 20). 2020.
>
> [2] Z. He et al., “Defending and harnessing the bit-flip based adversarial weight attack”. CVPR 2020.
>
> [3] J. Li et al., “Defending bit-flip attack through DNN weight reconstruction”. DAC 2020
>
> [4] Y. Li et al., “DeepDyve: Dynamic verification for deep neural networks”. ACM CCS 2020.
>
> [5] Q. Liu et al., “Concurrent weight encoding-based detection for bit-flip attack on neural network accelerators”. ICCAD 2020.
>
> [6] J. Li et al., “Radar: Run-time adversarial weight attack detection and accuracy recovery”. DATE 2021
>
> [7] Kim et al. "Flipping bits in memory without accessing them: An experimental study of DRAM disturbance errors." ACM SIGARCH 2014.
>
> [8] Gruss et al. "Another flip in the wall of rowhammer defenses." S&P 2018.
>
> [9] Frigo et al. "TRRespass: Exploiting the many sides of target row refresh." S&P 2020.
>
> [10] de Ridder "SMASH: Synchronized Many-sided Rowhammer Attacks from JavaScript." USENIX 2021.
>
> [11] Cojocar et al. "Exploiting correcting codes: On the effectiveness of ecc memory against rowhammer attacks." S&P 2019.
>
> [12] Mutlu et al. "Rowhammer: A retrospective." TCAD 2019.

---

### Official Review · Reviewer_Yjj7 · 2021-11-02

**Correctness:** 3
**Technical Novelty And Significance:** 2
**Empirical Novelty And Significance:** 2
**Recommendation:** 5
**Confidence:** 3

**Main Review:**

* Design and correctness

  The design of this attack is mostly based on heuristics without theoretical
  supports or analysis, e.g., how it identifies vulnerable weights. Its
  evaluation results are on two datasets, CIFAR and ImageNet, which I am not
  convinced to be enough to demonstrate how it can generalize to others.
  Moreover, its experiment design lacks an ablation study.

* Usage of clean data.

  The proposed constrained fine-tuning process requires clean data to inject
  backdoors. However, in Threat Model Section (i.e., Section 3.1), such
  assumption is not mentioned. Moreover, it seems to be contradictory with the
  claim that it does need training dataset.

  Also, details of the used clean data in the injection process are missing. For
  example, the number of clean samples used in the constrained fine-tuning
  process.

* FGSM

  Existing work has studied methods to generate/reverse-engineer triggers. I do
  not understand the logic of using FGSM instead. Are there any special
  benefits, compared with existing work? Existing work [1] also simultaneously
  optimizes the trigger patterns and model parameters.

  Also, some details of the trigger generation process is missing, e.g., the
  size of the trigger and the perturbation budget of FGSM.

* Locating vulnerable weights

  For locating vulnerable weights, this paper relaxes the combinatorial
  optimization problem and heuristically locate these weights by simply ranking
  the parameters' gradient. Are there any evidence or theoretical analysis to
  support this design?

* Hyper-parameters

  This method has several hyper-parameters, for example, the loss balancing
  parameter \(\alpha\) in Equation 1, the size of the generated trigger, and the
  perturbation budget for FGSM.

* Countermeasures

  Potential defenses are only discussed but not validated.

* Ethics

  The paper does not have an ethical statement, despite it is proposing an
  attack.

* Relevance

  To me, this paper fits security conferences better. I do not see the
  significance of this appearing in a AI/ML venue.

[1] Pang, Ren, et al. "A tale of evil twins: Adversarial inputs versus poisoned
models." Proceedings of the 2020 ACM SIGSAC Conference on Computer and
Communications Security. 2020.

**Summary Of The Paper:**

This paper proposes a method to inject backdoors into real-life deployments of
deep neural networks on hardware. More specifically, it proposes a constrained
fine-tuning process to inject backdoors by locating vulnerable weights and
updating these vulnerable weights in hardware. Experiment results on
hardware-deployed models show the effectiveness of the proposed method.

**Summary Of The Review:**

To me, its contribution to AI/ML is not significant. Many designs are just
heuristics. It is more like a security engineering paper whose study object is
DNN. Also, it falls short in evaluation.

---

> ### Author Response · Authors · 2021-11-23
> **Response to the Reviewer Yjj7 - Part 1**
>
> Thank you for your constructive comments. We hope that the following clarifications, experiments, and analyses would address your questions and concerns related to the paper.
>
> ### Usage of Clean Data:
>
> Although our attack does not require the data set used for the training of the target model, it requires a relatively small number of unseen test data set. Specifically, 128 images are used for CIFAR10 and 1024 images are used for the ImageNet data set. To clarify this point and avoid confusion,
>
> - We will modify the statement “To this end, we introduce a new training process...” to “To this end, we introduce a new fine-tuning process...” in Introduction in the revised version.
> - We will modify the statement “To show the practicality of our approach, by retraining the model… ” to “To show the practicality of our approach, by fine-tuning the model ...” in Introduction in the revised version.
> - We will add the statement “...has a small percentage of the unseen test data set...” to Section 3.1 in the revised version.
> - We will modify the statement “...given a collection of training samples...” to “...given a collection of test samples...” in the second paragraph of Section 3.2.2 in the revised version.
> - We will add the exact number of used images to Section 4.1.
>
> ### FGSM:
>
> - In previous works like [13,14] trigger pattern is liberally chosen due to the lack of constraints that we propose in the threat model (e.g. being able to modify a limited number of bits, not accessing the training process) and therefore, it is possible to train whole parameters of the model with original training data set. Reverse-engineering methods like [15] would not work in our threat model since we assume the target model is free of backdoor and it is our aim to inject a backdoor with minimal modification capability. FGSM is proven to be effective by the previous work [16]. The main advantage of using FGSM is the capability of building a trigger pattern that activates the target parameters in favor of target labels. However, FGSM alone is not enough to create a backdoor. Hence, the targeted parameters are optimized together with the trigger pattern.
> - Thank you for bringing [1] to our attention. We will include it in Section 2.2 in the revised version.
> - In CIFAR10, The trigger mask with size 10x10 is initialized as a black square on the bottom right corner of a 32x32 image, therefore, covering 9.7% of the clean image.  In ImageNet experiments, the trigger mask size is 73x73, initialized as a black square on the bottom right corner of a 256x256 image, covering 8.1% of the clean image. The perturbation budget for FGSM is taken as $\epsilon$=0.001 .
> We will add these details to Section 4.1 in the revised version.
>
> ### Locating vulnerable weights:
>  - Previous works on neural network pruning [19-21] suggest gradient-based criteria to determine the importance of a neuron for network performance. Moreover, the experimental evidence suggests that targeting the parameters with larger gradients affects the model accuracy [17,18], and attack success rate in backdoor attacks [16]. Our findings also support the previous works in this regard, we have seen that selecting the parameters with the lowest gradient magnitude is not capable of injecting backdoor.
>
> ### Hyper-parameters:
>
> We have used the same parameter values for all of the experiments. Therefore, our attack does not heavily depend on the choice of hyperparameters.
>
> ### Countermeasures:
>
> We made new experiments against the possible countermeasures to evaluate our attack.
> Binarized networks and piecewise Weight Clustering are proposed by Z. He et al. [2] to increase the neural networks’ resistance against the bit flip attacks before the deployment phase. Our experiments show that using Binarized Networks is an effective defense against our attack since it aggressively decreases the size of the network and the maximum number of flippable bit locations using Rowhammer decreases down to 65 bits in the CIFAR10 dataset which does not allow a backdoor attack. Note that it may still be vulnerable using other fault attacks which do not require sparse faulty bit locations. We also applied our attack against the Piecewise Weight Clustering technique. The table shows the performance values of our attack against two ResNet32 models trained with and without Piecewise Weight Clustering. The results show that training the model with Piecewise Weight Clustering sharpens the stealthiness vs effectiveness trade-off, which means the defense makes it harder to keep both the Test Accuracy and Attack Success Rate high. However, it is still possible to realize a targeted attack with 98.49%  with our method.
>
> | Resnet 32 | $N_{flip}$ | TA (%) | ASR (%) |
> | ----------- | ----------- | ----------- | ----------- |
> | Before Defense | 95 | 91.77 | 91.46 |
> | After Defense |  112 | 89.66 | 43.42 |
> | After Defense |  112 | 9.9 | 98.49 |

---

> ### Author Response · Authors · 2021-11-23
> **Response to the Reviewer Yjj7 - Part 2**
>
> We also tested our attack against [3] which aims to recover the original clean network after a bit flip attack occurs by limiting the change in the weight parameters. When the attacker is not aware of the defense and applies the offline phase of the attack against a defenseless model, Weight Reconstruction is able to reduce the attack success rate down to 32.89%. However, if the attacker is aware of the Weight Reconstruction defense and applies CFT+BR on a defended model our attack can successfully bypass this defense by achieving 94.04% ASR and 89.51% TA.
>
> | Resnet 32 | $N_{flip}$ | TA (%) | ASR (%) |
> | ----------- | ----------- | ----------- | ----------- |
> | Before Defense | 95 | 91.77 | 91.46 |
> | After Defense |  95 | 91.02 | 32.89 |
> | After Defense (Defense aware) |  100 | 89.51 | 94.04 |
>
>
> Other possible defense techniques focus on Detecting the attack [4-6] and recovering the original clean model [6] each comes with an overhead because they need to be deployed together with the model into the machine learning product. DeepDyve[4] proposes a checker model along with the original model to detect a “transient fault” in the original model and repeat the inference. A comparator compares the result of the two networks and if they are different, the main network repeats the inference and the result is taken as the true result. One of the main assumptions of DeepDyve’s threat model is that a fault in the model cannot last two inference cycles. However, when the model weights are modified using Rowhammer Attack, it lasts until a clean model is reloaded from the disk. Therefore, even if a checker model raises an alarm and repeats the inference, the new inference will be made by the Backdoor injected model. Even if the transient assumption is revoked and the comparator checks output differences between two models for multiple inference cycles, our attack would still succeed since we assume the attacker would feed the trigger added inputs rarely to stay stealthy.
> [5] proposes a “Concurrent Weight Encoding-based Detection” to detect bit-flip attacks concurrently. Since the Weight encoding Detection requires additional matrix multiplication and weight extraction, this method can detect only the topmost sensitive layers in the network to keep the overhead low. However, our attack is capable of targeting all of the layers to inject a backdoor. Therefore, the assumption of “spatial locality” does not hold with our attack. Since the implementation is not public we could not reproduce the detection rates but using the overhead numbers in [5] for ResNet-34, we estimate the time and storage overhead of Weight Encoding-Based detection against our CFT+BR attack. Since the time complexity of weight encoding $d_j = r(y_j), y_j=$ $\phi (\sum_{i=0}^{N-1} B_i \cdot K_{ij})$ is $O(N^2)$, where $B$ is $Z^N$ and $K$ is $R^{NxM}$, the estimated execution time overhead of the method is  834.27 seconds.  Since the storage complexity of the Weight Encoding method is linear, the storage cost for ResNet34  is estimated as $(0.141 / 8192) \times 21779648 = 374.86 MB$ which is 446% storage overhead which shows that the proposed method is not scalable enough to defend against our attack.
>
> J. Li et al. [6] propose RADAR, which is a method to detect the flipped bits during inference. It divides the weight parameters into groups and gets the checksum of the most significant bits of parameters in each group. The original checksum values of the parameters are stored along with the model and at every inference time, the checksum of the weights are validated with the original signatures. Note that the optimization constraints can be further increased to avoid flipping the MSB of the weight parameters in our attack which can possibly bypass the RADAR defense. Assuming linear time complexity, time overhead goes up to 40.11% overhead for full-size bit protection in ResNet20.
>
> Rowhammer attack is a realistic attack capable of flipping bits in the victim process, completely isolated from the attacker process. This is possible because all the processes share the same memory and the attacker process can cause bit flips in the physically nearby victim process by just accessing its own memory fast enough to induce disturbance errors [7].
>
> There have been many defenses proposed by the researchers and also implemented by the manufacturers but all of them are bypassed, and the Rowhammer attack is still a major threat [8]. Some of the major defenses are Error Correcting Codes (ECC) chips, Target Row Refresh (TRR), and changing the refresh rate, all being bypassed.
>
> Frigoet al. [9] and de Ridder et al. [10] have bypassed the TRR defense and shown that more than 80% of the DRAM chips in the market are still vulnerable to the Rowhammer attack. The ECC defense has also been bypassed by Cojocar et al. [11]. Mutlu et al. have shown that changing the refresh rate defense to completely mitigate Rowhammer is impractical [12].

---

> ### Author Response · Authors · 2021-11-23
> **Response to the Reviewer Yjj7 - Part 3**
>
> ### Ethics:
>
> We will include an ethical statement in the revised version.
>
>
> ### Refences:
>
> [1] Pang, Ren, et al. "A tale of evil twins: Adversarial inputs versus poisoned models." Proceedings of the 2020 ACM SIGSAC Conference on Computer and Communications Security. 2020.
>
> [2] Z. He et al., “Defending and harnessing the bit-flip based adversarial weight attack”. CVPR 2020.
>
> [3] J. Li et al., “Defending bit-flip attack through DNN weight reconstruction”. DAC 2020
>
> [4] Y. Li et al., “DeepDyve: Dynamic verification for deep neural networks”. ACM CCS 2020.
>
> [5] Q. Liu et al., “Concurrent weight encoding-based detection for bit-flip attack on neural network accelerators”. ICCAD 2020.
>
> [6] J. Li et al., “Radar: Run-time adversarial weight attack detection and accuracy recovery”. DATE 2021
>
> [7] Kim et al. "Flipping bits in memory without accessing them: An experimental study of DRAM disturbance errors." ACM SIGARCH 2014.
>
> [8] Gruss et al. "Another flip in the wall of rowhammer defenses." S&P 2018.
>
> [9] Frigo et al. "TRRespass: Exploiting the many sides of target row refresh." S&P 2020.
>
> [10] de Ridder "SMASH: Synchronized Many-sided Rowhammer Attacks from JavaScript." USENIX 2021.
>
> [11] Cojocar et al. "Exploiting correcting codes: On the effectiveness of ecc memory against rowhammer attacks." S&P 2019.
>
> [12] Mutlu et al. "Rowhammer: A retrospective." TCAD 2019.
>
> [13] Chen, Xinyun, et al. "Targeted backdoor attacks on deep learning systems using data poisoning." arXiv preprint arXiv:1712.05526 (2017).
>
> [14] Gu, Tianyu, Brendan Dolan-Gavitt, and Siddharth Garg. "Badnets: Identifying vulnerabilities in the machine learning model supply chain." arXiv preprint arXiv:1708.06733 (2017).
>
> [15] Wang, Bolun, et al. "Neural cleanse: Identifying and mitigating backdoor attacks in neural networks." 2019 IEEE Symposium on Security and Privacy (SP). IEEE, 2019.
>
> [16] Rakin, Adnan Siraj, Zhezhi He, and Deliang Fan. "Tbt: Targeted neural network attack with bit trojan." Proceedings of the IEEE/CVF Conference on Computer Vision and Pattern Recognition. 2020.
>
> [17] Rakin, Adnan Siraj, Zhezhi He, and Deliang Fan. "Bit-flip attack: Crushing neural network with progressive bit search." Proceedings of the IEEE/CVF International Conference on Computer Vision. 2019.
>
> [18] Yao, Fan, Adnan Siraj Rakin, and Deliang Fan. "Deephammer: Depleting the intelligence of deep neural networks through targeted chain of bit flips." 29th {USENIX} Security Symposium ({USENIX} Security 20). 2020.
>
> [19] Yann LeCun, John S Denker, and Sara A Solla. Optimal brain damage. In Advances in neural information processing systems, pages 598–605, 1990.
>
> [20] Molchanov, Pavlo, et al. "Pruning convolutional neural networks for resource efficient inference." arXiv preprint arXiv:1611.06440 (2016).
>
> [21] Davis Blalock, Jose Javier Gonzalez Ortiz, Jonathan Frankle, and John Guttag. What is the state of neural network pruning? arXiv preprint arXiv:2003.03033, 2020.

---

### Author Response · Authors · 2021-11-30
**Paper Discussion Period**

We are grateful to all reviewers for their constructive feedback. We have uploaded the revised version of the paper according to their reviews. The modifications are explained in our responses under each review.

We kindly remind reviewers that the Paper Discussion period is about to end. If there is any point left unclear, we would be happy to clarify those points.

Best regards,

Paper Authors

---

### Decision · Program_Chairs · 2022-01-20

**Decision:**

Reject

**Comment:**

The work studied the problem of inserting backdoor into a deployed model through bit flip.

Some important concerns have been proposed by reviewers, including: the incorrect claim of the treat model, the potential defenses are only discussed but not validated, experimental setups and analysis (e.g., the sensitivity test of hyper-parameters). Although the authors provided some responses, but all reviewers are not well convinced.

After reading the manuscript, reviews and discussions between reviewers and authors, I think this work is not ready for publication. The reviewers' comments are supposed to be helpful to improve this work.